# Sharp Impossibility Results for Hypergraph Testing

**Jiashun Jin**
Carnegie Mellon University
jiashun@stat.cmu.edu

**Zheng Tracy Ke**
Harvard University
zke@fas.harvard.edu

**Jiajun Liang**
Purdue University
liangjj@purdue.edu

## Abstract

In a broad Degree-Corrected Mixed-Membership (DCMM) setting, we test whether a non-uniform hypergraph has only one community or has multiple communities. Since both the null and alternative hypotheses have many unknown parameters, the challenge is, given an alternative, how to identify the null that is *hardest* to separate from the alternative. We approach this by proposing a *degree matching strategy* where the main idea is leveraging the theory for tensor scaling to create a least favorable pair of hypotheses. We present a result on standard minimax lower bound theory and a result on *Region of Impossibility* (which is more informative than the minimax lower bound). We show that our lower bounds are tight by introducing a new test that attains the lower bound up to a logarithmic factor. We also discuss the case where the hypergraphs may have mixed-memberships.

## 1 Introduction

The hypergraph is a useful representation of social relationships beyond pairwise interactions [5, 11]. For example, the co-authorship hypergraph is often used to analyze the co-authorship topology of authors, and it provides more information than a co-authorship graph (where an $m$-author paper is treated as an $m$-clique). The community detection on a hypergraph [10] is a problem of great interest (communities in a hypergraph are clusters of nodes that have more hyperedges within than across). It has many applications in social network analysis [15] and machine learning [1, 17, 18, 23]. We are interested in the problem of *global testing*, where we test whether the hypergraph has one community or multiple communities. It has applications in measuring co-authorship and citation diversity [12] and discovering non-obvious social groups and patterns [4]. It is also useful for understanding other problems such as community detection and change-point detection in dynamic hypergraphs.

For instructional purpose only, we start with the 3-uniform hypergraphs (i.e., each hyperedge consists of 3 nodes), but our results cover both higher-order hypergraphs and non-uniform hypergraphs. Let $\mathcal{A}$ be the adjacency tensor of a uniform and symmetric order-3 hypergraph with $n$ nodes, where

$$\mathcal{A}_{i_1 i_2 i_3} \text{ equals } 1 \text{ if } i_1, i_2, i_3 \text{ share a hyperedge and equals } 0 \text{ otherwise}, \qquad (1.1)$$

for $1 \le i_1, i_2, i_3$ (distinct) $\le n$. Since the tensor is symmetric, $\mathcal{A}_{i_1 i_2 i_3} = \mathcal{A}_{j_1 j_2 j_3}$ for two sets of indices $\{i_1, i_2, i_3\}$ and $\{j_1, j_2, j_3\}$ if one is a permutation of the other. We do not consider self hyperedges, hence, $\mathcal{A}_{i_1 i_2 i_3} = 0$ whenever $i_1, i_2, i_3$ are non-distinct.

Real world hypergraphs have several noteworthy features. First, there may be severe degree heterogeneity (i.e., the degree of one node is many times higher than that of another). Second, the overall sparsity levels may vary significantly from one hypergraph to another. Last, a node may have mixed-memberships across multiple communities (i.e., nonzero weights on more than one community). To accommodate these features, we adopt the *Degree-Corrected Mixed-Membership* (tensor-DCMM) model. The notations below are frequently used in tensor analysis.

**Definition 1.1.** *(matricization, slicing, and slice aggregation). Let $\mathcal{A}$ be a 3-symmetric tensor of $n$-dimension. First, we call the $n \times n^2$ matrix $A$ the matricization of $\mathcal{A}$, defined by $A_{i,j+n(k-1)} = \mathcal{A}_{ijk}$,*

35th Conference on Neural Information Processing Systems (NeurIPS 2021).

$1 \le i, j, k \le n$. Second, for $1 \le k \le n$, we use $\mathcal{A}_{::k}$ to denote the $n \times n$ matrix whose row-$i$-and-column-$j$ is $\mathcal{A}_{ijk}$, $1 \le i, j \le n$, and call it the $k$-th slice of $\mathcal{A}$. Last, for any $n \times 1$ vector $x$, we use $(\mathcal{A}x)$ to denote the matrix $\sum_{k=1}^{n} x_k \mathcal{A}_{::k}$, which is an aggregation of the slices.

Now, suppose the tensor has $K$ perceivable communities. Let $\mathcal{P}$ be a symmetric (3-uniform) tensor of $K$-dimension that models the community structure, let $\theta = (\theta_1, \theta_2, \ldots, \theta_n)'$ be positive parameters that model the degree heterogeneity of nodes, and $\pi_1, \pi_2, \ldots, \pi_n$ be $K$-dimensional membership vectors where $\pi_i(k) = $ the weight node $i$ puts on community $k$, $1 \le k \le K$. We assume for all $1 \le i_1, i_2, i_3 \le n$ that are three distinct indicies, $\mathcal{A}_{i_1 i_2 i_3}$ are independent Bernoulli random variables with $\mathbb{P}(\mathcal{A}_{i_1 i_2 i_3} = 1) = \theta_{i_1} \theta_{i_2} \theta_{i_3} \sum_{k_1, k_2, k_3 = 1}^{K} \mathcal{P}_{k_1, k_2, k_3} \pi_{i_1}(k_1) \pi_{i_2}(k_2) \pi_{i_3}(k_3)$, where by Definition 1.1, the right hand side equals to $\theta_{i_1} \theta_{i_2} \theta_{i_3} \pi'_{i_1} (\mathcal{P} \pi_{i_3}) \pi_{i_2}$. Introduce a non-stochastic 3-uniform tensor $\mathcal{Q}$ of $n$-dimension where $\mathcal{Q}_{i_1 i_2 i_3} = \theta_{i_1} \theta_{i_2} \theta_{i_3} \pi'_{i_1} (\mathcal{P} \pi_{i_3}) \pi_{i_2}$ for $1 \le i_1, i_2, i_3 \le n$. Let $\mathrm{diag}(\mathcal{Q})$ be the tensor with the same size of $\mathcal{Q}$ where $(\mathrm{diag}(\mathcal{Q}))_{i_1 i_2 i_3} = \mathcal{Q}_{i_1 i_2 i_3}$ if $i_1, i_2, i_3$ are non-distinct, and $(\mathrm{diag}(\mathcal{Q}))_{i_1 i_2 i_3} = 0$ otherwise. It follows that

$$\mathbb{E}(\mathcal{A}) = \mathcal{Q} - \mathrm{diag}(\mathcal{Q}), \qquad \text{where} \quad \mathcal{Q}_{i_1 i_2 i_3} = \theta_{i_1} \theta_{i_2} \theta_{i_3} \pi'_{i_1} (\mathcal{P} \pi_{i_3}) \pi_{i_2}. \qquad (1.2)$$

For identifiability, let $P \in \mathbb{R}^{K, K^2}$ be the matricization of $\mathcal{P}$ (see Definition 1.1). We assume

$$\mathrm{rank}(P) = K, \quad \text{and} \quad \mathcal{P}_{iii} = 1 \text{ for all } 1 \le i \le K. \qquad (1.3)$$

**Definition 1.2.** *We call (1.1)-(1.3) the tensor-DCMM model for 3-uniform hypergraphs. We call $\mathcal{Q}$ and $\mathcal{P}$ the Bernoulli probability tensor and community structure tensor for DCMM, respectively.*

Later in Section 3, we introduce the *non-uniform tensor-DCMM* as a more sophisticated model. In tensor-DCMM, if we require all $\pi_i$ to be degenerate (i.e., one entry is 1, all other entries are 0), then tensor-DCMM reduces to the *Degree-Corrected Block Model (tensor-DCBM)* [15]. If we further require $\theta_1 = \ldots = \theta_n$ (but the second condition in (1.3) can be removed), then tensor-DCBM further reduces to the *Stochastic Block Model* (tensor-SBM) [9]. For simplicity, we may drop "tensor" in these terms if there is no confusion. The global testing problem above is then to test

$$H_0 : K = 1 \qquad \text{vs.} \qquad H_1 : K > 1. \qquad (1.4)$$

Our primary goals are (a) to find a sharp information lower bound for 3-uniform DCMM, and especially, to fully characterize the lower bound by *a simple quantity to be discovered*, and (b) extend the results to more sophisticated non-uniform hypergraphse (see Section 3). A good understanding of the problem greatly helps us understand the fundamental limits of many other problems (e.g., community detection, determining the number of communities $K$, dynamic hypergraphs). For example, for community detection (e.g., [10]), we either assume $K$ as known or estimate it first. Note that in parameter regions where we cannot tell whether $K = 1$ or $K > 1$, we cannot estimate $K$ consistently, so we cannot have consistent community detection either. Therefore, a lower bound for global testing is always a valid lower bound for estimating $K$ and for community detection.

To facilitate the lower bound study, we frequently adopt a *Random Mixed-Membership (RMM)* model. Introduce a subset $V_0 = \{x \in \mathbb{R}^K : x_k \ge 0, \sum_{k=1}^{K} x_k = 1\}$, and let $F$ be a $K$-variate distribution with support contained in $V_0$. We assume

$$\pi_i \overset{iid}{\sim} F; \qquad (\text{let } h = \mathbb{E}_F[\pi_i]). \qquad (1.5)$$

Moreover, let $V_0^* = \{e_1, e_2, \ldots, e_K\} \subset V_0$, where $e_k$ is the $k$-th basis vector of $\mathbb{R}^K$. Similarly, RMM-DCMM reduces to RMM-DCBM if we require $\mathrm{supp}(F) = V_0^*$, and reduces to RMM-SBM if we further require $\theta_1 = \theta_2 = \ldots = \theta_n$ (but the second condition in (1.3) can be removed). Let $\theta = (\theta_1, \theta_2, \ldots, \theta_n)'$. We allow $(\theta, \mathcal{P}, h, F)$ to vary with $n$ to cover a variety of settings where we allow for severe degree heterogeneity, mixed-memberships, flexible sparsity levels, and weak signals.

**Example 1** *(2-parameter SBM [2, 22, 19])*. This model is a special case of DCMM, where $\theta_1 = \ldots = \theta_n = \alpha_n$ (no degree heterogeneity), all $\pi_i$ are degenerate (no mixed membership), and $\mathcal{P}_{ijk} = 1$ if $i = j = k$ and $\mathcal{P}_{ijk} = \rho_n$ otherwise, for two parameters $(\alpha_n, \rho_n)$. Also, Lin et al. [20] studied a 3-parameter SBM, which is the same as above except that they assume a different form of $\mathcal{P}$, where $\mathcal{P}_{ijk}$ equals to 1, $\rho_n$, or $\tau_n$ if $i, j, k$ take 1, 2, or 3 distinct values, respectively, for three parameters $(\alpha_n, \rho_n, \tau_n)$. Compared to DCMM, these models are much narrower: they do not accommodate severe degree heterogeneity or mixed-memberships, and $\mathcal{P}$ is parametrized by 2 or 3 different values.

How to derive a sharp lower bound for global testing (and especially, to identify a *simple quantity* that fully characterizes the lower bound) in our setting is a rather challenging problem. Our model is a non-uniform hypergraph model (see Section 3), which consists of hypergraphs of order $2, 3, \ldots, M$, and each layer consists of many unknown parameters $(\theta, \mathcal{P}, h, F)$. Existing works on lower bounds have been focused on uniform hypergraphs, and non-uniform hypergraphs are much less studied. Even for uniform hypergraphs, existing works have been focused on the the special SBM as in Example 1, not the more general DCMM model. For example, Yuan et al. [22] derived the lower bounds for global testing with the 2-parameter SBM, focusing on the extremely sparse case. Ahn et al. [2] provided lower bound results for exactly recovering the communities (see Liang et al. [19] and Kim et al. [16] for related settings) with a similar model. Lin et al. [20] and Chien et al. [7] used the 3-parameter SBM in Example 1 for study of the lower bounds for community detection. While these papers are very interesting, their lower bounds are characterized by only 2 or 3 parameters (i.e., $\alpha_n, \rho_n, \tau_n$) assumed in their models. For the much broader tensor-DCMM model considered here, we have many parameters $\theta, \mathcal{P}, h, F$ ($\theta$ is an $n$-vector and $\mathcal{P}$ is a tensor), and how to extend existing results to the tensor-DCMM setting here is a challenging problem.

**Our contributions**. The main challenge in lower bound study is that, since each DCMM model (no matter what $K$ is) has a large number of unknown parameters, so given an alternative hypothesis ($K > 1$), it is hard to identify the null hypothesis ($K = 1$) that is *most difficult to distinguish* from the alternative hypothesis. As our main contribution, we approach this by proposing a *degree matching strategy*, where for any given DCMM model with $K > 1$, we pair it with a DCMM model with $K = 1$ in a way so that for *each node*, the expected degree under the null matches with that under the alternative. This way, it is hard to separate the two hypotheses by a naive degree-based statistic. We show (a) the degree matching is always possible by using a tensor scaling technique [3, 8], and (b) the pair of hypotheses we construct this way lead to sharp results on lower bounds. See Section 2.

We have the following results. Consider the 3-uniform hypergraph first. We first present the standard minimax theory. Define a class of RMM-DCMM models with $K > 1$ and $\mu_2^2 \|\theta\|^2 \|\theta\|_1 \to 0$ ($\mu_2$ is the second singular value of $P$). In this class, we can find an RMM-DCMM model (the alternative), and pair it with a DCMM model with $K = 1$ (the null), such that the $\chi^2$-divergence between the pair converges to 0 as $n \to \infty$. Therefore, in this class, there *exists an alternative model* that is asymptotically inseparable from the null.

The standard minimax theory only claims that *there exists an alternative model* within a specified class that is inseparable from the null. It is desirable to show a much stronger result where *for any alternative in the class*, we can pair it with a null so that the $\chi^2$-divergence of the pair goes to 0 as $n \to \infty$. In detail, we show that in the parameter space $(\theta, \mathcal{P}, h, F)$ of RMM-DCBM, there is a *Region of Impossibility* defined by $\mu_2^2 \|\theta\|^2 \|\theta\|_1 \to 0$; for *any alternative* in Region of Impossibility, we can pair it with a null such that the $\chi^2$-divergence between the pair goes 0 as $n \to \infty$. Compared with existing results on minimax lower bounds, these results are more informative and theoretically more satisfactory. The proof is also different from the proof of minimax lower bounds: we have used the tensor scaling theory [3, 8] and the "degree matching strategy" aforementioned. We also extend such results to the broader RMM-DCMM case (Section 2.3) and discuss some major differences on the Region of Impossibility between DCBM and the more restrictive SBM (Section 2.4).

Next, we generalize the results to higher-order and non-uniform hypergraphs. Fix $M \geq 2$ and consider a non-uniform hypergraph (e.g., see [10]) that consists of $m$-uniform hypergraphs for all $m = 2, \ldots, M$, each following a DCMM model with individual $(\theta^{(m)}, \mathcal{P}^{(m)})$ but the common $\pi_1, \ldots, \pi_n$. Let $\ell_m = \|\theta^{(m)}\|_1^{m-2} \|\theta^{(m)}\|^2 (\mu_2^{(m)})^2$ (to be defined in Section 3). We show that (a) for the $m$-uniform hypergraph case, the Region of Impossibility for the hypothesis testing (1.4) is fully characterized by the condition of $\ell_m \to 0$, and (b) for the non-uniform hypergraph case, the Region of Impossibility for the hypothesis testing (1.4) is fully characterized by the condition of $\max_{2 \leq m \leq M} \{\ell_m\} \to 0$.

Last, we show that our lower bounds are tight. Consider the non-uniform hypergraph above. We propose a new test statistic and show that the sum of Type I error and Type II error $\to 0$ if $\max_{2 \leq m \leq M} \{\ell_m\} \geq \log(n)^{1+\delta}$ for a constant $\delta > 0$ (taking $\delta = 0.1$ will work). Therefore, except for a logarithmic factor here, our lower bounds are tight.

In summary, existing results on lower bounds are largely focused on more restrictive settings (e.g., uniform hypergraphs without degree heterogeneity or mixed membership). We provide sharp lower bounds for a much broader setting. Our study is highly non-trivial because we need (i) a novel *degree*

*matching strategy* to construct least favorable hypothesis pairs, (ii) to identify a simple quantity that is able to fully characterize the lower bounds, (iii) delicate analysis of the $\chi^2$-divergence between the null and alternative, (iv) a carefully designed test that leads to tight upper bounds.

## 2 Sharp Lower Bounds for $3$-Uniform Hypergraphs

For notational simplicity, we focus on 3-uniform hypergraphs in this section. The study of higher-order and non-uniform hypergraphs is deferred to Section 3. Consider a (3-uniform) DCMM model. Recall that $h = \mathbb{E}_F[\pi_i]$ and that $P \in \mathbb{R}^{K, K^2}$ is the matricization of the tensor $\mathcal{P}$. In this paper, we use $C > 0$ as a generic constant which may vary from occurrence to occurrence. We assume

$$\|P\| \leq C, \ \theta_{\max} \equiv \max\{\theta_1, \ldots, \theta_n\} \leq C, \ \max_{1 \leq k \leq K}\{h_k\} \leq C \min_{1 \leq k \leq K}\{h_k\}, \ \|\theta\|^2\|\theta\|_1 \to \infty. \quad (2.6)$$

The first condition is mild, because the model identifiability already requires that all diagonal entries of $\mathcal{P}$ are 1. The second one is also mild (note that while the largest possible value of $\theta_{\max}$ is $O(1)$, $\theta_{\max}$ is allowed to tend to 0 relatively fast). The third one assumes that the community memberships are balanced, which is also mild. For the last condition, we will see soon that if $\|\theta\|^2\|\theta\|_1 \to 0$, then the signal is so weak that successful global testing is impossible, so this condition is also mild.

### 2.1 Standard minimax lower bounds (RMM-DCMM)

We start with the least favorable configuration. The goal is to find a pair of models (a null model with $K = 1$, and an alternative model with $K > 1$) which are hard to distinguish from each other. We use the following degree-matching technique: we choose the pair in a way so that for each node, the expected degree under the null matches with that under the alternative, approximately. The idea is, if the degrees are not matching, then we may separate the two hypotheses by a simple degree-based statistic, so we should not expect the $\chi^2$-divergence between the pair to be small.

In detail, consider a pair of models, a DCMM model with $K = 1$ and an RMM-DCMM model with $K > 1$, where for all $1 \leq i_1, i_2, i_3 \leq n$, the Bernoulli probability tensors $\mathcal{Q}$ and $\mathcal{Q}^*$ satisfy

$$\mathcal{Q}_{i_1 i_2 i_3} = \theta_{i_1}\theta_{i_2}\theta_{i_3}, \qquad \mathcal{Q}^*_{i_1 i_2 i_3} = \theta^*_{i_1}\theta^*_{i_2}\theta^*_{i_3} \cdot \pi'_{i_1}(\mathcal{P}\pi_{i_3})\pi_{i_2}; \quad (2.7)$$

here we recall that $(\mathcal{P}\pi_{i_3})$ is a $K \times K$ matrix (see Definition 1.1). We call the two models *the null* and *the alternative*, respectively. In (2.7), the community structure tensor $\mathcal{P}$ is as in (1.2), and $\pi_i$ and $h = \mathbb{E}_F[\pi_i]$ are as in (1.5). Recall that $\theta = (\theta_1, \theta_2, \ldots, \theta_n)'$. Similarly, let $\theta^* = (\theta^*_1, \theta^*_2, \ldots, \theta^*_n)'$. Fix $1 \leq i_1 \leq n$. By definitions and elementary statistics, the leading term of the expected degree of node $i_1$, conditional on its own membership $\pi_{i_1}$, under the null and alternative are

$$\theta_{i_1}\|\theta\|^2_1 \ \text{ and } \ \theta^*_{i_1} \cdot \|\theta^*\|^2_1 \cdot \pi'_{i_1}a, \ \text{ respectively, where } a = (\mathcal{P}h)h \in \mathbb{R}^K. \quad (2.8)$$

For least favorable construction, we choose $(\mathcal{P}, h)$ in a way so that

$$a = (\mathcal{P}h)h = c_0^3\mathbf{1}_K, \qquad \text{for a scalar } c_0 > 0. \quad (2.9)$$

For broadness, we allow $c_0$ to depend on $n$. There are many $(\mathcal{P}, h)$ that satisfy (2.9). For example, in the 2-parameter SBM model as in Example 1 with $h = (1/K, \ldots, 1/K)'$ and $a_k = (1/K^2)[1 + (K^2 - 1)\rho_n]$ for $1 \leq k \leq K$, (2.9) is satisfied with $c_0^3 = (1/K^2)[1 + (K^2 - 1)\rho_n]$. Moreover, we choose $\theta^*_i$ in the alternative model such that

$$\theta^*_i = (1/c_0)\theta_i, \qquad 1 \leq i \leq n. \quad (2.10)$$

Now, by (2.8)-(2.10), for all $1 \leq i_1 \leq n$, $\theta^*_{i_1}\|\theta^*\|^2_1 \cdot \pi'_{i_1}a = \theta^*_{i_1}\|\theta^*\|^2_1 \cdot c_0^3 = \theta_{i_1}\|\theta\|^2_1$. Therefore, for each node, the expected degree under the alternative matches that under the null (at least in the leading term), making it hard to separate the null and alternative by any naive degree-based statistics. Only when such a degree-matching holds, we can hope two models are asymptotically indistinguishable. This is the key for our least favorable configuration. Recall that $P \in \mathbb{R}^{K, K^2}$ is the matricization of the community tensor $\mathcal{P}$ and $\mu_k$ is the $k$-th largest singular value of $P$.

**Theorem 2.1** (Least favorable configuration). *Fix $K > 1$ and consider a pair of models, a null and an alternative with $K$ communities, given in (2.7), where (2.9)-(2.10) hold. Assume (2.6) holds and*

$\|\theta\|_1 \|\theta\|^2 \mu_2^2 = o(1)$. *As $n \to \infty$, the $\chi^2$-divergence [1] between the pair tends to $0$. Therefore, the two models are asymptotically indistinguishable: for any test, the sum of Type I and Type II errors is no smaller than $1 + o(1)$.*

**Remark** *(How degree matching affects the $\chi^2$-divergence).* Intuitively speaking, the $\chi^2$-divergence has many terms, each being the sum (or a function) of terms in a Taylor expansion. We can roughly call these terms the first-order term, second-order term, and so on. When the expected node degrees are not matched between the null and the alternative, the first-order term dominates, and correspondingly, a degree-based $\chi^2$-test may have power; see Section 2.4. When the expected degrees are matched, the first-order term vanishes as we have hoped, and the degree-based $\chi^2$-test loses power. As a result, the second-order term in the $\chi^2$-divergence now dominates and gives the sharp lower bound.

The above least favorable configuration gives rises to the standard minimax theorem. Fix $K \geq 1$ and consider a DCMM model (where $\pi_i$ are non-random). Introduce a vector $g \in \mathbb{R}^K$ by $g_k = (1/\|\theta\|_1) \sum_{i=1}^{n} \theta_i \pi_i(k)$, $1 \leq k \leq K$. For a constant $0 < c_0 < 1$, and two positive sequences $\{\alpha_n\}_{n=1}^{\infty}$ and $\{\beta_n\}_{n=1}^{\infty}$, we define a class of DCMM models by

$$\mathcal{M}_n(K, c_0, \alpha_n, \beta_n) = \left\{ \begin{array}{ll} (\theta, \Pi, P): & \|P\| \leq c_0, \; \theta_{\max} \leq c_0, \; \max_{1 \leq k \leq K} g_k \leq c_0^{-1} \min_{1 \leq k \leq K} g_k, \\ & \|\theta\|_1 \|\theta\|^2 \geq 1/\beta_n, \; \|\theta\|_1 \|\theta\|^2 \mu_2^2 \leq \alpha_n \end{array} \right\}.$$

Here, we assume $\max_{1 \leq k \leq K}\{g_k\} \leq C \min_{1 \leq k \leq K}\{g_k\}$, so the tensor is balanced. This is similar to the third condition in (2.6), except that $\pi_i$'s are random there. For the null case, $K = P = \pi_i = 1$, and the above defines a class of $\theta$, which we write for short by $\mathcal{M}_n(1, c_0, \alpha_n, \beta_n) = \mathcal{M}_n^*(\beta_n)$. The following theorem is proved in the supplement.

**Theorem 2.2** (Minimax lower bound). *Fix $K \geq 2$, a constant $c_0 > 0$, and any sequences $\{\alpha_n\}_{n=1}^{\infty}$ and $\{\beta_n\}_{n=1}^{\infty}$ where $\alpha_n = o(1)$ and $\beta_n = o(1)$. As $n \to \infty$, $\inf_{\psi} \{\sup_{\theta \in \mathcal{M}_n^*(\beta_n)} \mathbb{P}(\psi = 1) + \sup_{(\theta, \Pi, P) \in \mathcal{M}_n(K, c_0, \alpha_n, \beta_n)} \mathbb{P}(\psi = 0)\} = 1 - o(1)$, where the infimum is over all possible tests $\psi$.*

## 2.2 Region of Impossibility for RMM-DCBM

The standard minimax theorem in Section 2.1 only says that in the class of all alternative models with $\|\theta\|^2 \|\theta\|_1 \mu_2^2 \to 0$, *there exists one* where we can pair it with a null so the pair are asymptotically inseparable. A much more satisfactory result is to show that, for *any alternative* in the same class, we can pair it with a null such that the pair are asymptotically inseparable. In this section, we prove this for the RMM-DCBM case (mixed-memberships are not allowed). The discussion for the RMM-DCMM (mixed-memberships allowed) is in Section 2.3 and the supplement.

Consider again a pair of models, a DCBM null model and an RMM-DCBM model with $K > 1$, where the Bernoulli probability tensors are $\mathcal{Q}$ and $\mathcal{Q}^*$, respectively. We assume for all $1 \leq i_1, i_2, i_3 \leq n$,

$$\mathcal{Q}_{i_1 i_2 i_3} = \theta_{i_1} \theta_{i_2} \theta_{i_3}, \qquad \mathcal{Q}_{i_1 i_2 i_3}^* = \theta_{i_1}^* \theta_{i_2}^* \theta_{i_3}^* \pi_{i_1}' (\mathcal{P}\pi_{i_3}) \pi_{i_2}. \tag{2.11}$$

Here, the community structure tensor $\mathcal{P}$ is as in (1.2), $\pi_i$ and $h = \mathbb{E}_F[\pi_i]$ are as in (1.5), and $\text{supp}(F) = \{e_1, \ldots, e_K\}$. Similarly, the goal is degree-matching: for any $(\theta, \mathcal{P}, h, F)$, we construct $\theta^*$ in a way so that for each node, the expected degrees under the null and the alternative match with each other approximately. Recall that in Section 2.2, in order to have a desired degree matching, it is crucial to pick an alternative model where the $(\mathcal{P}, h)$ satisfies $a \equiv (\mathcal{P}h)h = c_0^3 \mathbf{1}_K$ for some scalar $c_0 > 0$; once this holds, we have the desired degree matching by taking $\theta^* = (1/c_0)\theta$. Unfortunately, for general $(\mathcal{P}, h)$, we don't have $a \equiv (\mathcal{P}h)h \propto \mathbf{1}_K$, so we should not expect to have the desired degree matching by taking $\theta^* \propto \theta$. In short, for our purpose here, the approach in Section 2.1 no longer works, and we must find a new approach to constructing the model pair.

Our proposal is as follows. For any diagonal matrix $D = \text{diag}(d_1, \ldots, d_K)$ with $d_k > 0$, $1 \leq k \leq K$, define a $K$-dimensional vector $a^D$ by ($\mathcal{P}^D$ is a 3-tensor in dimension $K$)

$$a^D = (\mathcal{P}^D h)h, \qquad \text{with} \quad \mathcal{P}_{k_1 k_2 k_3}^D = d_{k_1} d_{k_2} d_{k_3} \mathcal{P}_{k_1 k_2 k_3}, \; 1 \leq k_1, k_2, k_3 \leq K.$$

We aim to select a matrix $D$ such that $a^D = \mathbf{1}_K$. The next lemma states that such $D$ always exists and is unique. It leverages classic results on tensor scaling (e.g., [3, 8]) and is proved in the supplement.

---

[1] The $\chi^2$-divergence between two models, $f_0(\mathcal{A})$ and $f_1(\mathcal{A})$, is defined as $\int [(f_0(\mathcal{A}) - f_1(\mathcal{A}))^2 / f_0(\mathcal{A})]d\mathcal{A}$. In our setting, the alternative model $f_1(\mathcal{A})$ alone involves an integral over the distribution of $\pi_i$ in (1.5)

**Lemma 2.1.** *Fix $K > 1$ and let $\mathcal{P}$, $h$, $D$, and $a^D$ be as above. Suppose $\min\{h_1, h_2, \ldots, h_K\} \geq C$. There exists a unique diagonal matrix $D = \mathrm{diag}(d_1, d_2, \ldots, d_K)$ such that $a^D = \mathbf{1}_K$.*

Now, given $(\theta, \mathcal{P}, h, F)$ and the two models in (2.11), let $D = \mathrm{diag}(d_1, d_2, \ldots, d_K)$ be as in Lemma 2.1. Moreover, in (2.11), we choose $\theta_i^*$ as follows:

$$\theta_i^* = d_k \theta_i, \quad \text{if node } i \text{ belongs to community } k. \tag{2.12}$$

Combining it with (2.11), we have $\mathcal{Q}_{i_1 i_2 i_3}^* = \theta_{i_1}^* \theta_{i_2}^* \theta_{i_3}^* \cdot \pi_{i_1}'(\mathcal{P}\pi_{i_3})\pi_{i_2} = \theta_{i_1}\theta_{i_2}\theta_{i_3} \cdot \pi_{i_1}'(\mathcal{P}^D\pi_{i_3})\pi_{i_2}$. By similar calculations as in (2.8), for $1 \leq i \leq n$, in the null and the alternative, the leading terms of the expected degrees of node $i$ are

$$\theta_i \|\theta\|_1 \quad \text{and} \quad \theta_i(\pi_i' a^D)\|\theta\|_1^2, \quad \text{respectively, where } a^D = (\mathcal{P}^D h)h.$$

By Lemma 2.1, $a^D = \mathbf{1}_K$. Hence, for each node, the expected degrees match under the null and alternative, so it is hard to separate two models by a naive degree-based statistic. Recall that $P$ is the matricization of $\mathcal{P}$ and $\mu_k$ is the $k$-th singular value of $P$. Theorem 2.3 is proved in the supplement.

**Theorem 2.3** (Impossibility for RMM-DCBM)**.** *Fix $K > 1$. For any given $(\theta, \mathcal{P}, h, F)$, consider a pair of models, a null and an alternative with $K$ communities, as in (2.11), where $\theta_i^*$ are given by (2.12) with the matrix $D$ as in Lemma 2.1. Suppose (2.6) holds and $\|\theta\|_1 \|\theta\|^2 \mu_2^2 = o(1)$. As $n \to \infty$, the $\chi^2$-divergence between the pair tends to $0$. Therefore, the two models are asymptotically indistinguishable: for any test, the sum of Type I and Type II errors is no smaller than $1 + o(1)$.*

**Region of Possibility, Region of Impossibility, and tightness**. In the parameter space $(\theta, \mathcal{P}, h, F)$ for DCBM, we call the region prescribed by $\|\theta\|_1 \|\theta\|^2 \mu_2^2 \to 0$ the *Region of Impossibility*: by Theorem 2.3, for any model in this region, we can pair it with a null so that the pair are asymptotically inseparable. We call the region prescribed by $\|\theta\|_1 \|\theta\|^2 \mu_2^2 / \log^{1.1}(n) \to \infty$ the *Region of Possibility*: for any alternative model in this region, there is a method that can separate it from *any null* with asymptotically full power (this follows from Theorem 3.2 as a special case). Comparing Region of Impossibility and Region of Possibility, except for a $\log(n)$ term, our lower bounds are tight.

## 2.3 Region of Impossibility for RMM-DCMM

The discussion for RMM-DCMM is similar, so for reasons of space, we leave it to Section A of the supplement. In that section, we present a similar theorem for RMM-DCMM, where the hypergraphs may have mixed-memberships. The proofs are largely similar, where again the key is to construct a pair of null and alternative using the degree matching strategy. Since the model is more complicated than RMM-DCBM, we need an extra (but mild) condition. See details therein.

## 2.4 Major differences on Region of Impossibility for the more restrictive RMM-SBM

For an alternative RMM-DCBM, we can always pair it with a null using tensor scaling technique: for each node, the expected degrees under the null and the alternative match with each other. For the more restrictive RMM-SBM (where $\theta_1 = \ldots = \theta_n$), such a degree matching is not always possible: A null SBM has only 1 parameter, so we have insufficient flexibility in choosing the null for degree matching. A consequence is that a naive degree-based test may have non-trivial power for SBM.

Consider a pair of models, where the Bernoulli probability tensors $\mathcal{Q}$ and $\mathcal{Q}^*$ under two hypotheses are such that, for $1 \leq i_1, i_2, i_3 \leq n$,

$$\mathcal{Q}_{i_1 i_2 i_3} = \alpha_n, \qquad \mathcal{Q}_{i_1 i_2 i_3}^* = \pi_{i_1}'(\mathcal{P}\pi_{i_3})\pi_{i_2}. \tag{2.13}$$

Same as before, $\pi_i$ are *iid* generated from a distribution $F$ on $\{e_1, e_2, \ldots, e_K\}$ with $h = \mathbb{E}_F[\pi_i]$. Let $\hat{\alpha}_n = (n(n-1)(n-2))^{-1}\mathbf{1}_n'(\mathcal{A}\mathbf{1}_n)\mathbf{1}_n$, $\eta = (1/2)(\mathcal{A}\mathbf{1}_n)\mathbf{1}_n$, and $\bar{\eta}$ be the mean of $\eta_1, \eta_2, \ldots, \eta_K$. Consider the centered-$\chi^2$-statistic

$$\psi_n = (2n)^{-1/2} \sum_{1 \leq i \leq n} \left[ (\eta_i - \bar{\eta})^2 / \left( \binom{n-1}{2} \hat{\alpha}_n (1 - \hat{\alpha}_n) \right) - 1 \right].$$

**Lemma 2.2.** *Consider the global testing problem (1.4) under the SBM model (2.13) for $H_0$ and $H_1$, respectively. Let $\tilde{\alpha}_n = \mathbb{E}[\hat{\alpha}_n]$, $h = (1/n)\sum_{i=1}^n \pi_i$ and $\Sigma = \frac{1}{n}\sum_{i=1}^n (\pi_i - h)(\pi_i - h)'$. Let $\lambda_{K-1}(\Sigma)$ be the $(K-1)$-th largest eigenvalue (in magnitude) of $\Sigma$. Assume $\alpha_n \leq c_0$, $\max_{1 \leq i,j,k \leq K}\{\mathcal{P}_{ijk}\} \leq c_0$, $n^2 \alpha_n \to \infty$, and $n^2 \tilde{\alpha}_n \to \infty$. Also, assume $\min\{h_1, h_2, \ldots, h_K\} \geq C$ and $\lambda_{K-1}(\Sigma) \geq C$. Write $\delta_n = \|\tilde{\alpha}_n^{-1}(I_K - H_K)(\mathcal{P}h)h\|$, where $H_K = (1/K)\mathbf{1}_K\mathbf{1}_K'$ and $I_K$ is the identity matrix of the same size. As $n \to \infty$, $\psi_n \to N(0,1)$ if $H_0$ holds, and $\psi_n \to \infty$ if $H_1$ holds and $n^{3/2}\tilde{\alpha}_n^{1/2}\delta_n^2 \to \infty$*

By Lemma 2.2, the power of the $\chi^2$-test hinges on $\delta_n$

$$\delta_n = \|\widetilde{\alpha}_n^{-1}(I_K - H_K)(\mathcal{P}h)h\|.$$

Note that $\delta_n = 0$ if and only if $(\mathcal{P}h)h \propto \mathbf{1}_K$. We call the cases of $(\mathcal{P}h)h \propto \mathbf{1}_K$ and $(\mathcal{P}h)h \not\propto \mathbf{1}_K$ the *symmetric* case and the *asymmetric* case, respectively. For symmetric SBM, $\delta_n = 0$, and we do not expect the $\chi^2$-test to have power. However, for asymmetric SBM, the $\chi^2$-test may have non-trivial power, implying a potential shift of the lower bound.

**Example 2**. Consider an SBM setting where we either have $K = 1$ (null) or $K = 2$ (alternative). Also, when $K = 2$, we assume $h = (a, 1 - a)'$ for some $0 < a < 1$, $\mathcal{P}_{ijk}$ is equal to $\rho_0$ if $i = j = k$ and $\rho_1$ otherwise. Suppose $n^2 \rho_0 \to \infty$ and $\rho_1/\rho_0 \to 1$. This can be viewed as a special DCBM with $\theta_i \equiv \rho_0^{1/3}$ and off-diagonals of $\mathcal{P}$ being $\rho_1/\rho_0$. In this case, we have $\|\theta\|_1\|\theta\|^2 \asymp n^2\rho_0$, $\tau_n \equiv \|\theta\|_1\|\theta\|^2\mu_2^2 \asymp n^2\rho_0^{-1}(\rho_1 - \rho_0)^2$, and $n^{3/2}\widetilde{\alpha}_n\delta_n^2 \asymp (n\rho_0)^{-1/2}(2a - 1)^2\tau_n$. In the symmetric case where $a = 1/2$, the $\chi^2$-test is powerless, and the Region of Impossibility is given by $\tau_n \to 0$ (same as in the DCBM case). In the asymmetric case where $|a - 1/2| \geq c_0$, by direct calculations, we have that even when $\tau_n \to 0$, we may have $n^{3/2}\widetilde{\alpha}_n^{1/2}\delta_n^2 \to \infty$ (so $\chi^2$-test has asymptotically full power). Here, the interesting range of $\rho_0$ is $(n^{-2}, 1)$, so we may have $(n\rho_0)^{-1/2} \to \infty$. Therefore, for the asymmetric case, the Region of Impossibility is different from that of DCBM.

In most lower bound results for SBM [2, 7, 20, 22], they focused on the symmetric case $(\mathcal{P}h)h \propto \mathbf{1}_K$. Our lower bound restricted to symmetric SBM agrees with those in the literature. The asymmetric case $(\mathcal{P}h)h \not\propto \mathbf{1}_K$ is less studied, except for [14, 21] which focused on the network setting ($m = 2$). We discover: (i) the Region of Impossibility for symmetric SBM is similar to that of DCBM (see Example 2), and (ii) the Region of Impossibility for asymmetric SBM is quite different from that of DCBM. This is because DCBM is much broader than SBM, where the problem of global testing is much harder, and a naive degree-based test statistic may lose power.

## 3  Sharp lower bound for non-uniform hypergraphs

Section 2 discusses lower bounds for uniform 3-hypergraph. We now first extend the results to more general non-uniform hypergraphs, and then present a tight upper bound. Note that our results include those for $m$-uniform hypergraphs as special cases (see the paragraph behind Theorem 3.1). The notation below is useful:

**Definition 3.1.** *Given an order-$m$ tensor $\mathcal{M}$ in dimension $K$ and vectors $b_1, b_2, \ldots, b_m \in \mathbb{R}^K$, let $[\mathcal{M}; b_1, \cdots, b_m]$ denote the summation $\sum_{1 \leq k_1, k_2, \ldots, k_m \leq K}[\mathcal{M}_{k_1 k_2 \ldots k_m} b_1(k_1) b_2(k_2) \cdots b_m(k_m)]$.*

Fix $M \geq 2$. Consider a general non-uniform hypergraph that consists of $m$-uniform hypergraphs for all $2 \leq m \leq M$. Fixing $2 \leq m \leq M$, let $\mathcal{A}^{(m)}$ be the adjacency tensor of the order-$m$ hypergraph (i.e., $\mathcal{A}^{(m)}_{i_1 i_2 \cdots i_m} = 1$ if $\{i_1, i_2, \ldots, i_m\}$ is a hyper-edge and 0 otherwise). As before, we model $\mathcal{A}^{(m)}$ with the tensor-DCMM model. Let $\mathcal{P}^{(m)}$ be a symmetric order-$m$ tensor in dimension $K$, and let $\theta^{(m)} = (\theta_1^{(m)}, \theta_2^{(m)}, \ldots, \theta_n^{(m)})'$ be a positive vector of degree parameters. Let $\pi_1, \pi_2, \ldots, \pi_n$ be the $K$-dimensional membership vectors (which do not depend on $m$). We assume $\{\mathcal{A}^{(m)}_{i_1 i_2 \cdots i_m}\}_{1 \leq i_1 < i_2 < \ldots < i_m \leq n}$ are independent Bernoulli variables, where the Bernoulli probabilities are specified by the tensor $\mathcal{Q}^{(m)}$, given by

$$\mathcal{Q}^{(m)}_{i_1 i_2 \ldots i_m} = \theta_{i_1}^{(m)} \ldots \theta_{i_m}^{(m)} \times [\mathcal{P}^{(m)}; \pi_{i_1}, \cdots, \pi_{i_m}], \qquad 1 \leq i_1, i_2, \ldots, i_m \leq n. \tag{3.14}$$

Similar to (1.2), we have $\mathbb{E}[\mathcal{A}^{(m)}] = \mathcal{Q}^{(m)} - \text{diag}(\mathcal{Q}^{(m)})$. This extends the tensor-DCMM model to $m$-uniform hypergraphs for a general $m$. Finally, we denote the non-uniform hypergraph by $\mathcal{A}[M] \equiv \{\mathcal{A}^{(2)}, \mathcal{A}^{(3)}, \ldots, \mathcal{A}^{(M)}\}$.

**Definition 3.2.** *We say that $\mathcal{A}[M] = \{\mathcal{A}^{(2)}, \ldots, \mathcal{A}^{(M)}\}$ follows a (general non-uniform) tensor-DCMM model if $\{\mathcal{A}^{(m)}\}_{2 \leq m \leq M}$ are independent of each other and each $\mathcal{A}^{(m)}$ follows an $m$-uniform tensor-DCMM model as in (3.14), where $\pi_1, \pi_2, \ldots, \pi_n$ are shared by all $2 \leq m \leq M$.*

A similar model is introduced in [10], but is more restrictive for it assumes $\theta_1^{(m)} = \theta_2^{(m)} = \cdots = \theta_n^{(m)}$ for each $2 \leq m \leq M$. Note also the focus of [10] is on community detection, while the focus here is

on global testing. As argued before, since in parameter regions where we cannot tell whether $K = 1$ or $K > 1$, it is impossible to estimate $K$ and community labels consistently. Therefore, our lower bound is also a valid lower bound for estimating $K$ and for community detection.

We present the *Region of Impossibility* for testing $H_0$: $K = 1$ versus $H_1$: $K > 1$. Similarly as in Section 2.2, we focus on the special case of tensor-DCBM models (i.e., each $\pi_i$ is degenerate); the study of tensor-DCMM is similar. Fix $K > 1$. Consider a DCBM null model with probability tensors $\mathcal{Q}[M] = \{\mathcal{Q}^{(2)}, \ldots, \mathcal{Q}^{(M)}\}$ and an RMM-DCBM model with probability tensors $\mathcal{Q}^*[M] = \{\mathcal{Q}^{*(2)}, \ldots, \mathcal{Q}^{*(M)}\}$, where for every $2 \leq m \leq M$ and $1 \leq i_1, i_2, \ldots, i_m \leq n$,

$$\mathcal{Q}^{(m)}_{i_1, i_2, \ldots, i_m} = \theta^{(m)}_{i_1} \theta^{(m)}_{i_2} \cdots \theta^{(m)}_{i_m}, \tag{3.15}$$

$$\mathcal{Q}^{*(m)}_{i_1, i_2, \ldots, i_m} = \theta^{*(m)}_{i_1} \cdots \theta^{*(m)}_{i_m} \times [\mathcal{P}^{(m)}; \pi_{i_1}, \ldots, \pi_{i_m}], \qquad \pi_i \overset{iid}{\sim} F. \tag{3.16}$$

The support of $F$ is in $V_0^* = \{e_1, e_2, \ldots, e_K\}$. Let $h = \mathbb{E}_F[\pi_i]$ and suppose $\min\{h_1, \ldots, h_K\} \geq C$. In the supplemental material, we provide a lemma analogous to Lemma 2.1: For each $2 \leq m \leq M$, there exists a unique diagonal matrix $D^{(m)} = \mathrm{diag}(d_1^{(m)}, d_2^{(m)}, \ldots, d_K^{(m)})$ such that

$$\sum_{1 \leq i_2, \ldots, i_m \leq K} d_{i_1}^{(m)} \cdot \mathcal{P}^{(m)}_{i_1 \cdots i_m} \cdot (d_{i_2}^{(m)} h_{i_2}) \cdots (d_{i_m}^{(m)} h_{i_m}) = 1, \quad \text{for every } 1 \leq i_1 \leq K.$$

Its proof leverages the Sinkhorn theorems for higher-order tensors [3, 8]. We choose $\theta^{*(m)}$ in (3.16) by

$$\theta_i^{*(m)} = d_k^{(m)} \theta_i^{(m)}, \quad \text{if node } i \text{ belongs to community } k. \tag{3.17}$$

This is analogous to the degree matching strategy in (2.12), and it is conducted for each $m$ separately. Let $P^{(m)} \in \mathbb{R}^{K \times K^{m-1}}$ be the matricization of $\mathcal{P}^{(m)}$ and let $\mu_2^{(m)}$ be the second singular value of $P^{(m)}$. For short, let

$$\ell_m = \|\theta^{(m)}\|_1^{m-2} \|\theta^{(m)}\|^2 (\mu_2^{(m)})^2.$$

Theorem 3.1 is proved in the supplement.

**Theorem 3.1** (Impossibility for non-uniform RMM-DCBM). *Fix $K > 1$ and $M \geq 2$. For any given $(h, F)$ and $\{(\theta^{(m)}, \mathcal{P}^{(m)})\}_{2 \leq m \leq M}$, consider a pair of models, a null as in (3.15) and an alternative with $K$ communities as in (3.16), where $\{\theta_i^{*(m)}\}_{1 \leq i \leq n, 2 \leq m \leq M}$ are as in (3.17). Suppose $\|P^{(m)}\| \leq C$, $\max_{1 \leq i \leq n} \theta_i^{(m)} \leq C$, and $\min_{1 \leq k \leq K} h_k \geq C$. If $\max_{2 \leq m \leq M}\{\ell_m\} = o(1)$, then as $n \to \infty$, the $\chi^2$-divergence between the pair tends to 0.*

**Remark** *(Comparison with [22])*. Yuan et al. [22] gave a nice impossibility result for the 2-parameter symmetric SBM. Their model is a special case of our model where $\theta_i^{(m)} \equiv \alpha_n^{(m)}$ and $\mathcal{P}^{(m)}$ has equal off-diagonal entries $\rho^{(m)}$ (see Example 1 for $m = 3$). In this case, $|\mu_2^{(m)}| = |1 - \rho^{(m)}|$, and $\ell_m \asymp n^{m-1} \alpha_n^m (1 - \rho^{(m)})^2$. Hence, $\ell_m \to 0$ is equivalent to $n^{m-1} \alpha_n^{(m)} (1 - \rho^{(m)})^2 \to 0$, which matches with results in [22].

Below in Section 3.1, we show that the lower bounds are tight. Theorem 3.1 includes $m$-uniform hypergraphs as a special case (e.g., to apply the theorem to an $m_0$-uniform hypergraph, we set $\theta^{(m)}$ a zero vector for all $m \neq m_0$). For an $m$-uniform hypergraph, the Region of Impossibility is given by $\ell_m \to 0$, which is the same as that in Theorem 2.3 when $m = 3$. While many lower bound results are available for uniform hypergraphs [2, 7, 20, 22], non-uniform hypergraphs are less studied. Our lower bound in Theorem 3.1 leads to two notable discoveries: (i) the Regions of Possibility/Impossibility for $\mathcal{A}^{(m)}$ are fully characterized by the simple quantity of $\ell_m$; (ii) the Regions of Possibility/Impossibility for non-uniform hypergraphs are fully characterized by the simple quantity of $\max_{2 \leq m \leq M}\{\ell_m\}$.

### 3.1 Tightness of the lower bounds

We propose a test for the global testing problem (1.4) and show that it attains the lower bounds in Theorems 2.3 and Theorem 3.1. For simplicity, we only discuss this for DCBM with moderate degree heterogeneity but the tightness holds in much broader settings (e.g., DCMM).

For each $2 \leq m \leq M$, we first compute a vector $\eta^{(m)} \in \mathbb{R}^n$, which serves as an estimate of $\theta^{(m)}$ when the null hypothesis is true. Fix $m$. Given a positive vector $u \in \mathbb{R}^n$, define $L(u) \in \mathbb{R}^n$ by

$$L_i(u) = \frac{\sum_{i_2,\ldots,i_m(\text{distinct})} \mathcal{A}^{(m)}_{i\,i_2\ldots i_m} + \sum_{i_2,\ldots,i_m(\text{non-distinct})} u_i u_{i_2} \cdots u_{i_m}}{\left(\sum_{i_1,\ldots,i_m(\text{distinct})} \mathcal{A}^{(m)}_{i_1 i_2\ldots i_m} + \sum_{i_1,\ldots,i_m(\text{non-distinct})} u_{i_1} \cdots u_{i_m}\right)^{(m-1)/m}}. \quad (3.18)$$

Let $N_m = \lceil \frac{m-1}{2} \rceil$. Initialize at $u^{(0)} = \mathbf{0}_n$, compute $u^{(k)} = L(u^{(k-1)})$ iteratively for $k = 1, \ldots, N_m$, and output $u^{(N_m)}$ as $\eta^{(m)}$. We note that each $u_i^{(1)}$ is a simple function of node degrees, $1 \leq i \leq n$. For $m \in \{2, 3\}$, since $\eta^{(m)} = u^{(1)}$, we estimate $\theta^{(m)}$ directly from node degrees. However, for $m \geq 4$, $u^{(1)}$ is a biased estimator of $\theta^{(m)}$ under the null, and the bias is caused by $\text{diag}(\mathcal{Q}^{(m)})$. The iteration serves to reduce this bias. The required number of iterations depends on $m$ explicitly.

Next, we construct a statistic $Q_n^{(m)}$ to capture the difference between $\mathcal{A}^{(m)}$ and a rank-1 estimate of the Bernoulli probability tensor. Let $\mathcal{A}^{*(m)}$ be a tensor of the same size as $\mathcal{A}^{(m)}$, where $\mathcal{A}^{*(m)}_{i_1\ldots i_m} = \mathcal{A}^{(m)}_{i_1\ldots i_m} - \eta^{(m)}_{i_1} \cdots \eta^{(m)}_{i_m}$ for $1 \leq i_1, \ldots, i_m \leq n$. We say that $S = (S_1, S_2, \ldots, S_{m+1})$ is an $(m+1)$-partition of $\{1, 2, \ldots, n\}$ if $S_1, \ldots, S_{m+1}$ are disjoint and their union is $\{1, 2, \ldots, n\}$. Let $B$ be the set of all $(m+1)$-partitions. For each $S = (S_1, \ldots, S_{m+1}) \in B$ and $1 \leq k_1, \ldots, k_m \leq m+1$, let

$$Q_n^{(m)} = \max_{S=(S_1,\ldots,S_{m+1})\in B} \max_{1\leq k_1,\ldots,k_m\leq m+1} \left\{ \left| \sum_{i_1\in S_{k_1},\ldots,i_m\in S_{k_m}(\text{distinct})} \mathcal{A}^{*(m)}_{i_1\ldots i_m} \right| \right\}. \quad (3.19)$$

Finally, we combine $Q_n^{(2)}, \ldots, Q_n^{(M)}$. For each $m$, let $V_n^{(m)} = \binom{n}{m} \hat{\alpha}_n (1 - \hat{\alpha}_n)$, where $\hat{\alpha}_n = [(n-m)!/n!] \sum_{i_1,\ldots,i_m=1}^n \mathcal{A}^{(m)}_{i_1\ldots i_m}$. The test statistic is

$$\phi_n = \max_{2\leq m\leq M} \left\{ Q_n^{(m)} / [n \log(n)^{1.1} V_n^{(m)}]^{1/2} \right\}. \quad (3.20)$$

Recall that by Theorem 3.1, for any alternative with $\max_{2\leq m\leq M}\{\ell_m\} = o(1)$, we can pair it with a null so that the pair are asymptotically indistinguishable. The next theorem says that the proposed test statistic $\phi_n$ can successfully separate any alternative satisfying $\max_{2\leq m\leq M}\{\ell_m\} \gg \log^{1+\delta}(n)$ (for some $\delta > 0$; taking $\delta = 0.1$ is adequate) from the null. Therefore, except for a logarithmic factor, our lower bounds are tight. Recall that $\ell_m = \|\theta^{(m)}\|_1^{m-2} \|\theta^{(m)}\|^2 (\mu_2^{(m)})^2$. Let $\theta_{\max}^{(m)}$ and $\theta_{\min}^{(m)}$ be the maximum and minimum entry of the vector $\theta^{(m)}$, respectively.

**Theorem 3.2** (Tightness of lower bounds). *Consider the general tensor-DCBM model with $M \geq 2$. Let $h = \frac{1}{n}\sum_{i=1}^n \pi_i$, and let $P^{(m)} \in \mathbb{R}^{K,K^{m-1}}$ be the matricization of $\mathcal{P}^{(m)}$. Suppose $\|P^{(m)}\| \leq C$, $\max_{1\leq k\leq K}\{h_k\} \leq C \min_{1\leq k\leq K}\{h_k\}$, $\theta_{\max}^{(m)} \leq C\theta_{\min}^{(m)}$, $\theta_{\max}^{(m)} \leq c_0$ for a constant $c_0 < 1$, and $\|\theta^{(m)}\|_1^{m-2}\|\theta^{(m)}\|^2/\log(n) \to \infty$, for every $2 \leq m \leq M$. Then, $\phi_n \to 0$ in probability, if $H_0$ holds, and $\phi_n \to \infty$ in probability, if $H_1$ holds and $\max_{2\leq m\leq M}\{\ell_m\}/[\log^{1.1}(n)] \to \infty$.*

To speed up the computation of $\phi_n$ for large $n$, we introduce a proxy statistic by replacing the search over $B$ by a specific $\hat{B}$ as follows: conduct SVD on the matricization of $\mathcal{A}^{(m)}$, take the first $(m+1)$ left singular vectors, and apply the spectral algorithm [13] to partition nodes into $(m+1)$ groups. We use this partition $\hat{B}$ to replace the maximization over $B$ in (3.19), to get a proxy to $Q_n^{(m)}$; the test statistic $\phi_n$ is then defined similarly as in (3.20). As long as the hypergraph is not too sparse, this proxy works well and is computationally much faster.

# 4 Numerical study

We use simulated data to validate our theoretical results. Fix $(n, K, m) = (500, 2, 3)$. In **Experiment 1**, we consider the SBM model and verify that the Regions of Impossibility are different for symmetric and asymmetric SBM (see Section 2.4). Let $\theta_i = n^{-1/2}$ for $1 \leq i \leq n$, and $\mathcal{P}_{ijk} = 1$ if $i = j = k$ and $\mathcal{P}_{ijk} = 1/4$ otherwise. We consider a symmetric case where each communities have 250 nodes and an asymmetric case where two communities have 375 and 125 nodes, respectively. For each setting, we randomly generate the hypergraphs, apply the degree-based $\chi^2$-statistic $\psi_n$ in Section 2.4, and repeat for 500 times. The histograms of $\psi_n$ for two cases are on the left panel of the figure below

(green: symmetric alternative; red: asymmetric alternative; blue: density of $N(0,1)$). By Lemma 2.2, $\psi_n \approx N(0,1)$ in the null. Hence, the results suggest that $\psi_n$ is unable to distinguish the symmetric alternative from the null but can distinguish the asymmetric alternative from the null.

In **Experiment 2**, we consider DCBM and use the least-favorable configuration in Section 2.2 where the degree matching strategy is employed. We verify that a degree-based test such as $\psi_n$ indeed has no power. Let $(n, K, m)$ be the same as above. In the null, we let $\theta_i$ be iid drawn from $\mathrm{Pareto}(0.5, 5)$ and then re-normalize the vector of $\theta$ so that $n^{-2}\|\theta\|_1\|\theta\|^2 = c_n$, for $c_n = n^{-1}$; in the alternative, let $(\mathcal{P}^*, h)$ be the same as in the asymmetric case in Experiment 1 and generate $\theta_i^*$ is as in (2.12) ($D$ is obtained by treating $D(\mathcal{P}(Dh))Dh = 1_K$ as a nonlinear equation and apply the Matlab function `solve`). The histograms of $\psi_n$ under the null (green) and alternative (red) are shown on the middle panel of the figure below. The two histograms are inseparable from each other.

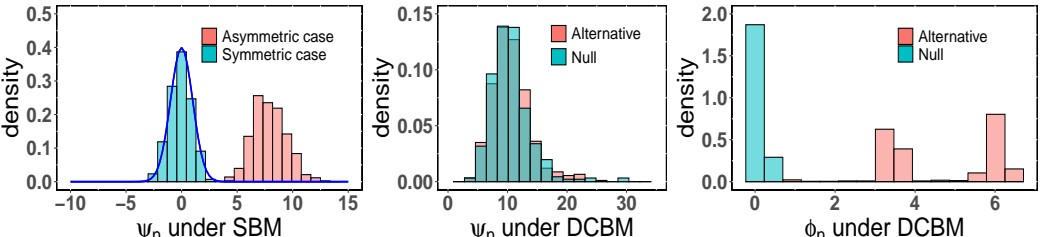

In **Experiment 3**, we study the test statistic $\phi_n$ proposed in Section 3.1. The simulation setting is the same as that in Experiment 2, except that $c_n = n^{-1/2}$. To save computing time, we use the proxy of $\phi_n$ by plugging in a $\hat{B}$ from spectral clustering (see Section 3.1). The histograms of the test statistic under the null (green) and alternative (red) are shown on the right panel of the figure above. We see that $\phi_n$ successfully distinguishes the alternative from the null. This validates our result in Theorem 3.2. The distribution of $\phi_n$ in the alternative has two modes, due to that the proxy $\hat{B}$ we plug in has two most frequent realizations. However, this does not affect the testing performance.

## 5  Conclusion

We consider the problem of global testing for non-uniform hypergraphs in a broad DCMM setting. Given an alternative, how to identify *the null that is hardest to separate from the alternative* is a challenging problem. We solve this by proposing a degree matching strategy, and use it to derive a tight lower bound by tensor scaling techniques and delicate analysis of the $\chi^2$-divergence. We discover that for an $m$-uniform hypergraph, the Regions of Impossibility/Possibility are governed by the simple quantity of $\ell_m = \|\theta^{(m)}\|_1^{m-2}\|\theta^{(m)}\|^2(\mu_2^{(m)})^2$ (and so those for a non-uniform hypergraph are governed by $\max_{2 \le m \le M}\{\ell_m\}$). We also propose a new test that attains the lower bounds, so our lower bounds are tight. For future work, we notice that the test in Section 3.1 is computationally expensive. It is desirable to find some fast algorithms that also achieve the lower bounds. The signed-cycle statistics [6, 14] are polynomial-time statistics that have shown appealing performance for network global testing. It is possible that these statistics can be generalized to hypergraphs to provide polynomial-time tests that are also theoretically optimal. We leave it to future study.

## Acknowledgments and Disclosure of Funding

The authors thank the anonymous reviewers for useful comments. J. Jin is supported in part by NSF Grant DMS-2015469. Z. Ke is supported in part by NSF CAREER Grant DMS-1943902.

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
