# OpenReview forum: "Sharp Impossibility Results for Hyper-graph Testing"
_NeurIPS.cc/2021/Conference — NeurIPS 2021 Poster_

### Official Review · Reviewer_V2Sb · 2021-07-14

**Rating:** 6
**Confidence:** 4

**Summary:**

The current paper explore information-theoretic results for testing the existence of community underlying an observed hypergraph. The nodes of hypergraph are degree hetergeneous extending homogeneous results in literature. Both uniform and non-uniform hypergraphs are considered.



**Ethics Review Area:**

["I don’t know"]

**Limitations And Societal Impact:**

There is a lack of comparison with existing results such as Yuan et al. (AoS, 2021) who explored testing limits under homogeneous setting, i.e., $\theta_1=\cdots=\theta_n$. Can the authors compare their results? For instance, setting all $\theta_i$'s to be equal, can they recover the results of   Yuan et al. (AoS, 2021) ?

Also, I am a bit confused by the definition of degree-corrected model which seems a bit different from literature. Can the authors explain why they consider such a different version?

**Main Review:**

The results are new, original, solving a basic problem in graphical model. The results are important, and proving deeper theoretical insights on the nature of the testing problem.

**Time Spent Reviewing:**

3

---

> ### Author Response · Authors · 2021-08-11
> **Point-by-point response to Reviewer V2Sb**
>
> Thanks for your valuable comments. Below is a point-to-point response (please also see our response to all reviewers above, where we clarify the motivations and practical side of the paper).
>
> Thanks, Yuan et al (2021) is a remarkable paper. We especially like the idea of using loose loop counts for testing, and the results on the power and limiting null there.  We have already included some comparison with the paper, and we would be happy to add more comparisons.
>
> As for the model, despite some notational differences, we are using the same model as in the literature (see more discussion below). For example, the setting of Yuan et al (2021) is a special case of our $m$-DCMM model if we let (a) $\theta_1 = \theta_2 = \ldots =  \theta_n = \alpha_n$, (b) all communities have the same size, and (c) ${\cal P}^{(m)}_{k_1k_2\cdots k_m} = 1$ if $k_1=k_2=\cdots =k_m$ and $\rho$  otherwise. Moreover, in this special case,   $|\mu_2^{(m)}| = |1-\rho|$ and $l_m=\\|\theta^{(m)}\\|_1^{m-2}\\|\theta^{(m)}\\|^2(\mu_2^{(m)})^2\asymp n^{m-1}\alpha_n^m(1-\rho)^2$, so $\ell_m \to 0$ is equivalent to that $n^{m-1}\alpha_n^m(1-\rho)^2\to 0$; the latter is the lower bound given by Yuan et al (2021). On the other hand, our paper is for the much broader non-uniform tensor model, and our results are more general. Even for the SBM case, note that our results show that symmetric SBM and asymmetric SBM have very different Region of Impossibility: degree-based $\chi^2$ is powerless in the former but may have power in the latter. Such insight was not discovered before.
>
> Our model is the same as that in the literature. One reason you may get confused is because we use a different notational system. For example, consider the network case without mixed-memberships. In this case,  ${\cal A}  = {\cal Q} - \mathrm{diag}(\Omega)$ and $\Omega = \Theta \Pi P \Pi' \Theta$, where $\Theta = \mathrm{diag}(\theta_1, \theta_2, \ldots, \theta_n)$ and $\Pi = [\pi_1, \pi_2, \ldots, \pi_n]'$. This is exactly the DCBM by Karrer and Newman (2010) for social networks.  Such a matrix form for $\Omega$ is highly preferred since the information bound critically depend on the eigenvalues of $\Omega$. Another reason that you may get confused is because we require $P$ to have diagonal entries (while SBM model does not have to). This is because DCBM is broader, so we need a constraint to make sure that $(\Theta, \Pi, P)$ are uniquely determined by $\Omega$. In fact,  for any positive numbers $d_1, d_2, \ldots, d_K$, let $D = \mathrm{diag}(d_1, d_2, \ldots, d_K)$ and $\widetilde{\Theta} = \mathrm{diag}(\tilde{\theta}_1, \tilde{\theta}_2, \ldots, \tilde{\theta}_n)$, where $\tilde{\theta}_i = d_k^{-1}  \theta_i$ if node $i$ belongs to community $k$. It follows that
> $$\Omega =  \Theta \Pi P \Pi' \Theta = \widetilde{\Theta} \Pi (D P D)  \Pi' \widetilde{\Theta}.$$
>
> Therefore, for any $(\Theta, \Pi, P)$ and $\Omega = \Theta \Pi P \Pi' \Theta$, we can always rewrite $\Omega = \widetilde{\Theta} \Pi \widetilde{P} \Pi'  \widetilde{\Theta}$ and  $\widetilde{P} = D P D$. But if we restrict $P$ to have diagonal entries, then $D$ has to be the identify matrix, so $(\Theta, \Pi, P)$ are uniquely defined.
>
> Last, note that in our response to all reviewers, we have presented an example where we consider an order-3 tensor with two equal-size communities where $\theta_1 = \theta_2 = \ldots = \theta_n = \alpha_n$ and where the two slices of the tensor ${\cal P}$ are
> \begin{equation}
> \left[
> \begin{array}{ll}
> .5 + \epsilon & .5 - \epsilon  \\\\
> .5 - \epsilon  & .5 + \epsilon  \\\\
> \end{array}
> \right], \qquad \mbox{and}
> \qquad \left[
> \begin{array}{ll}
> .5 - \epsilon & .5 + \epsilon   \\\\
> .5 + \epsilon  & .5 - \epsilon \\\\
> \end{array}
> \right],
> \end{equation}
> respectively. We conjecture that in this case, no polynomial-time test may achieve the lower bound. In case that you are not sure why this is true, let us note that the power of the test in Yuan et al (2021) depends on the quantity delta (defined in equation (10) of their paper), but delta = 0 in this particular example, so the test loses power. However, our proposed test (which is not polynomial-time) is optimal in this case. We hope this helps convince you that the setting we consider here is indeed much more complicated than that in Yuan et al (2021).

---

### Official Review · Reviewer_oDcv · 2021-07-18

**Rating:** 6
**Confidence:** 3

**Summary:**

This paper introduces and studies a hypergraph community detection model called RMM-DCMM. In this model, each vertex may have a different degree (degree-corrected) and may also belong to many different communities (mixed-membership).

A main takeaway from the paper is that possibility/impossibility of testing whether this complicated model has K = 1 or K > 1 communities boils down to the quantity ||\theta||^2 ||\theta||_1 \mu_2^2.

The main results are:
1) Region of impossibility result for RMM-DCBM (a subset of RMM-DCMM). This shows that given any:
	(a) degree-correction factors \theta for a 1-community model
	(b) tensor P representing connection strengths between subsets of the K communities,
	(c) distribution F over mixed membership vectors, supported on {e_1,...,e_K},
	if ||\theta||^2 ||\theta||_1 \mu_2^2 --> 0 (where \mu_2 is the second eigenvalue of the matricization of P), then there is a choice of degree-correction factors \theta^* such that the K-community model given  by (P, \theta^*, F) is information-theoretically indistinguishable from the 1-community model given by \theta

2) Region of possibility result for RMM-DCBM. For any K-community model (P,\theta,F) with K > 2, such that ||\theta||^2 ||\theta||_1 \mu_2^2 / log(n)^{1.1} --> \infty, there is a test (based on estimating the degree-correction vector \theta assuming that the model is null) to distinguish it from any 1-community model.

These results extend to RMM-DCMM under some additional conditions.

**Limitations And Societal Impact:**

Yes.

**Main Review:**

The paper appears technically correct, and the connection to tensor scaling is especially nice. Furthermore, it is nice that the paper boils down the study of a complicated testing problem to whether a simple quantity ||\theta||^2 ||\theta||_1 \mu_2^2 tends to 0 or to infinity. Finally, the RMM-DCMM model introduced by the paper is a natural community detection model.
Overall, this paper is of interest to researchers specializing in community detection, so it seems on-topic for the conference.

Nevertheless, I am on the fence about acceptance, since I think that the result may only be of interest to a very small group of attendees.
From a practical point of view, the testing algorithm to distinguish H0 (1 community) from H1 (more than one community), seems to depend heavily on the null distribution being Erdos-Renyi, and my best guess is that it will not be effective on real-world data (I could be convinced otherwise if the paper had experiments on real-world data). Furthermore, the testing algorithm is computationally inefficient, as it involves an exponentially-large sum. Because of this, the experiments implemented by the paper use an entirely different algorithm -- a standard spectral algorithm based on the matricization of the adjacency tensor.

Thus, the significance of the paper appears to be mainly theoretical. The main theoretical insight that I gleaned was that if the expected degrees of the K > 2-community model and the K = 1 community model match, then their information-theoretic distinguishability depends on the quantity ||\theta||^2 ||\theta||_1 \mu_2^2, and whether it tends to 0 or to infinity sufficiently quickly.


Minor comments

* "hypergraph" in the title should be capitalized for consistency
* typos: indicies, hypergraphse, "can not" --> "cannot", "the simply quantity"

* Line 127, \theta^{(m)}, P^{(m)} \mu_2^{(m)} have not yet been defined. Please either define them or add a remark that they will be defined later and are analogues of the 3-hypergraph case.

* Line 233, write log(n)^{1.1} instead of log(n)^{1+delta}

* Line 286, "[M; b1, · · · , bK]" should be "[M; b1, · · · , bm]"

* Lines 320, 321: (3.16) and (3.15) are flipped

**Time Spent Reviewing:**

4

---

> ### Author Response · Authors · 2021-08-11
> **Point-by-point response to Reviewer oDcV**
>
>
> Thank you for your reading and your careful summary. We are glad that you think our results are nice and our topic is relevant for the conference. Below is a point-to-point response (please also see our response to all reviewers above, where we discuss the motivation and
> practical side of our paper).
>
> 1.The null model.
>
> We must clarify that our null model is not the Erdor-Renyi model. In our null model, the probability for a hyper-edge $\{i,j,k\}$ is equal to $\theta_i\theta_j\theta_k$. Degree heterogeneity still exists in the null model.
>
> Your guess is correct that a test using the Erdos-Renyi null is usually not effective on real data applications. This is the motivation of our work: Degree-corrected models are preferred in real applications, but it is unclear to what extent its statistical limit differs from that for SBM in hypergraphs. Our study helps answer this question.
>
>
> 2.Whether the result is interesting to only a small group of attendees.
>
> We must clarify that community detection is a problem of interest to many researchers in social network data analysis (e.g., Peter Bickel and Bin Yu at UC-Berkeley, and Stephen Fienberg and Eric Xing at CMU used to work in this area). It is true that there are interests from the random graph and SBM community, but the main users of community detection algorithms are from machine learning, statistics, and social science. Network analysis is a well-accepted and indispensable area in machine learning, and many NIPS papers discuss network analysis.
>
> Our problem is also of interest to researchers who study tensor data. Hypergraph community detection is indeed a clustering problem with Bernoulli tensor data. It has close connections to clustering with Gaussian tensor data, an area that have a lot of publications in NeurIPS.
>
> For these reasons, we think our problem will be interesting to a broad group of attendees.
>
>
> 3.Practical relevance and real applications.
>
> Here is an example of why our theory is useful in practice. Suppose we are interested in knowing whether a group of NeurIPS authors have latent clusters. We can build a coauthorship hypergraph on these authors using their NeurIPS papers in recent years and test whether $K=1$. Suppose a data analyst suggests to compute the node degrees $d_1,d_2,\ldots,d_n$ and look at the chi-square statistic $\sum_{i=1}^n (d_i-\bar{d})^2$. Our theory immediately tells that this is a wrong approach, because any degree-based test will not have power.
>
> A concrete real application is in our ongoing work (see our response to all reviewers above; where we discuss a recent data set on the publication of statistics, which contains bibtex (title, author, keywords, abstract, references) and citation data of 83,331 papers published in 36 journals, spanning 41 years (1975-2015)).   We can apply our test to measure the collaboration diversity of statisticians. We build a coauthorship hypergraph. Then, for each author, we extract a personalized hypergraph (also called an ego-hypergraph) by restricting the node set to this author and his/her coauthors. We then apply the test on this personalized hypergraph to get a p-value. This p-value is used to measure the collaboration diversity of this author. We can also apply our test for hierarchical community detection. This is also discussed in our response to all reviewers above.
>
>
>
> 4.The algorithm in simulations.
>
> We clarify that the test used in our numerical experiments is not an "entirely different" algorithm. Our test can be re-expressed as $\max_{S} \{X(S)\}$, where the maximum is taken over all $(m+1)$-partitions of nodes and $X(\cdot)$ is an empirical utility function defined through lines 337-345 (given any $S$, if we follow the computation in lines 337-345, the resulting number is denoted by $X(S)$).
> The definition of $X(\cdot)$ is the core idea behind our test. In simulations, we replace the exhaustive search over $S$ by $\hat{S}$ from a spectral clustering algorithm, and we use $X(\hat{S})$ as the test statistic. Here, we use $\hat{S}$ to approximate the maximizer of  $X(\cdot)$. It is a proxy of our original test, not something "entirely different."
>
> There is a rationale for this proxy. Using the proof of Theorem 3.1 (and some regularity conditions), we can show that the maximizer of $X(\cdot)$ is a non-splitting partition, i.e., each group is equal to either a true community or the union of a few true communities, with high probability. This inspires the use of $\hat{S}$ from spectral clustering.
>
> Alternatively, we may replace the exhaustive search by a greedy search over sub-sampled $S$. We tried it numerically and found that the current proxy was better.
>
>
> 5.Our algorithmic contributions.
>
> There is no available test for the global testing under tensor-DCMM. The test we propose in Section 3.1 is entirely new. As a byproduct,  we also introduce a new iterative algorithm for estimating degree parameters in the null DCMM and a proxy test (see Point 4 above) that runs fast and works satisfactoriy simulations. These are all our algorithmic contributions. The proxy test has a great potential for real applications. Please see our response to all reviewers above (especially the last part, where we discuss three points on the algorithm).

---

> > ### Comment · Reviewer_oDcv · 2021-08-31
> > **response**
> >
> > Thank you for your detailed response. I realize that I did not understand the test that was used in the numerical experiments, and I recognize that this is an interesting contribution of this paper. Nevertheless, I am not planning on changing my score (which is Weak Accept), since I think that what I wrote about the practical significance of the result still stands.

---

> > > ### Author Response · Authors · 2021-09-01
> > > **Response to Reviewer oDcV**
> > >
> > > Thanks for your comments. We are glad that you think the test we use in simulations is an interesting contribution.
> > >
> > > In our point-by-point response (Point 3), we gave a specific example of how these results are practically useful, which we do not repeat here. We wish to take this opportunity to re-iterate our point: Lower bound studies are practically relevant, because they help differentiate optimal algorithms from non-optimal ones. In each NIPS conference, hundreds of new algorithms were proposed, but over the years, only a small fraction of them will pass the test of time. It is therefore desirable to develop algorithms that are optimal, which are more promising to pass the test of time. But to do that, we have to derive a sharp lower bound as a benchmark and this is particularly important since community detection on hypergraphs attracts increasing attention. For a problem, an optimal algorithm and a sharp lower bound are two sides of coin, and both are indispensable. From a practical viewpoint, advancements in each of two aspects are valuable.

---

### Official Review · Reviewer_jdLd · 2021-07-20

**Rating:** 5
**Confidence:** 3

**Summary:**

This paper considers the community detection problem in the hypergraph setting: determine whether there is a single community (null hypothesis) or K > 1 communities (alternate hypothesis). The main contribution is the development of lower bounds via bounding the \Chi^2-divergence, which is a standard method, although quite technically demanding in this setting due to the many parameters involved. Overall, this is a clear contribution to the theory literature on stochastic block models and community detection. It is a bit harder to judge its appropriateness for NeurIPS.

**Limitations And Societal Impact:**

I did not have any concerns on this front.

**Main Review:**

On the technical side, the main contribution of this paper is pushing through the challenging technical analysis for the \Chi^2-divergence to demonstrate a lower bound that is tight up to logarithmic factors. I believe this is a valuable theoretical contribution, that extends the community detection work from the random graph to the random hypergraph setting. The paper is a dense read, due to the need to capture all of the notation and technical details within the page limit. I did not check the proofs carefully, but the results seems correct.

It does not devote much space or effort to justifying where this form of hypergraph model might arise and where the obtained theoretical understanding might lead to some useful algorithms. This would have significantly strengthened the paper with respect to NeurIPS, and might give the broader audience an appreciation for why such these results might be needed as part of a larger body of work. As the paper currently stands, it is an interesting contribution to a seemingly narrow set of technical questions arising from the stochastic block model community, and might be a better fit in a venue that is more targeted towards that community. To be clear, I do not see a clear reason to reject this paper from NeurIPS but I do not see a compelling reason to include it either. The paper would have been much stronger with a sample application and a real dataset, to justify the complexity of the proposed hypergraph models. I completely agree that this is a challenging technical problem, but I would prefer to see how and why this level of complexity might be needed some day in a data-driven setting.

**Time Spent Reviewing:**

2

---

> ### Author Response · Authors · 2021-08-11
> **Point-by-point response to Reviewer jdLd**
>
>
> Thank you for your comments and your recognition of our theoretical contributions. Below is a point-to-point response (please also see our response to all reviewers above, where we explain the motivation and practical side of our paper).
>
> 1.Appropriateness for NeurIPS.
>
> We have clarified this point in the response to all reviewers (see above).
>
>
> 2.Where the model arises and why the results are relevant to practice.
>
> In social networks, it has been recognized that the stochastic block model is often too idealized for real networks, and the degree-corrected stochastic block model (DCBM) (Karrer and Newman, 2010) is the state of art. Our tensor-DCMM model is a natural extension of DCBM from graph networks to hypergraph networks. It was observed in [15] that this model fits real data very well and is more suitable than tensor-SBM. In short, our model follows the spirit of the famous DCBM for graph networks. Its practical relevance has been well justified (see the references in [15]).
>
>
> We now give an example for relevance of our problem in practice. Suppose we are interested in knowing whether a group of NeurIPS authors have latent clusters. We can build a coauthorship hypergraph on these authors using their NeurIPS papers in recent years and test whether $K=1$. Suppose a data analyst suggests to compute the node degrees $d_1,d_2,\ldots,d_n$ and look at the chi-square statistic $\sum_{i=1}^n (d_i-\bar{d})^2$. Our theory immediately tells that this is a wrong approach, because any degree-based test will not have power.
>
>
> 3.Sample application and real data set
>
> Note that in our response to all reviewers above, we have introduced a data set on the publication data of statisticians, and two application examples. For example, in our ongoing work, we can apply this hypergraph testing to measure the collaboration diversity of statisticians. We build a coauthorship hypergraph using publications in statistics journals. Then, for each author, we extract a personalized hypergraph (also called an ego-hypergraph) by restricting the node set to this author and his/her coauthors. We then apply the test on this personalized hypergraph to get a p-value. This p-value is used to measure the collaboration diversity of this author. We also refer to Gao and Lafferty (2017) for a similar application on coauthorship graphs. Also, see our response to all reviewers for how our problem of global  testing is related to hierarchical/recursive community detection (a problem of great interest in network and tensor analysis).

---

### Official Review · Reviewer_LQ2Q · 2021-07-27

**Rating:** 5
**Confidence:** 4

**Summary:**

This paper studies a binary hypothesis testing problem that considers whether a given hypergraph has one or more than one community. Under a general class of degree-corrected mixed-membership model for hypergraphs, it identifies conditions under which it is impossible for any test to distinguish between the null and alternative hypothesis. The above results, which are first derived for uniform hypergraphs, are then extended to non-uniform hyper-graphs by viewing a non-uniform hypergraph as a collection of uniform hypergraphs. The paper also proposes a statistical test that successfully solves the above-mentioned hypothesis testing problem under conditions that can get logarithmically close to the impossibility regime. Experiments on simulated data validate some of the theoretical findings.

**Limitations And Societal Impact:**

Yes

**Main Review:**

 1. This paper argues that it is the first to consider significantly more flexible models that support mixed-membership and degree-heterogeneity in hypergraphs. The key idea is degree matching, which allows construction of a difficult-to-distinguish alternative hypothesis. It would be better to present an intuitive explanation for the technical difficulties that arise due to the use of a richer model. What techniques have been borrowed from the existing literature and why are existing techniques not applicable in the current setting?

2. Why is $\pi_{i_1}$ still present while computing the expected degree of node i_1 under the alternative hypothesis? Under the alternative hypothesis, $\pi_{i_1}$ is a random quantity.

3. Degree matching aims at matching the expected degrees of nodes under the null and alternative model. Thus, it appears reasonable to argue that a degree-based test would fail in this case. However, to argue that "any" test would fail, the paper shows that $\chi^2$-divergence between the null and alternative distribution goes to zero asymptotically. An intuitive explanation for how one goes from arguing that expected degree matches to arguing that $\chi^2$-divergence is zero would be very helpful in making the paper more accessible.

4. Sections 2.4 and 3.1 are difficult to understand. Consider providing additional details and more intuition.


**Time Spent Reviewing:**

4

---

> ### Author Response · Authors · 2021-08-10
> **Point-by-point response to Reviewer LQ2Q**
>
> Thank you for your very helpful comments. Below is our point-by-point response (please also read our response to all reviewers above,
> where we explain the motivation and practical side of our paper).
>
>
> 1.The challenges and difficulties that arise due to the use of a richer mode:
>
> Compared with the study of tensor-SBM, we encounter 4 major challenges as follows:
>
> First, in lower bound studies, a key step is to figure out the "least favorable configuration" -- how to pair the null and alternative hypotheses so that testing is most difficult. A naive extension of the least favorable configuration from tensor-SBM to tensor-DCMM will suggest matching the degree parameters $\theta_1,\ldots,\theta_n$ of two hypotheses. Unfortunately, this fails to lead to a sharp lower bound. We find out that the correct approach is matching expected node degrees, not matching the degree parameters. These approaches happen to be the same for tensor-SBM but they are different for tensor-DCMM. This was not known before.
>
> Second, there is no existing approach for matching the expected node degrees (because this is not needed for tensor-SBM). We tackle it by borrowing the tensor scaling literature (generalization of Sinkhorn-type theorems from matrices to tensors) to propose a degree matching strategy. This is new and non-obvious.
>
> Third, the analysis of the $\chi^2$-divergence is challenging. Tensor-DCMM has many more parameters than tensor-SBM, and so the $\chi^2$-divergence is more complicated. It requires a lot of efforts to bound terms with different random patterns, and it is non-trivial to figure out that $\\|\theta\\|_1^{m-2}\\|\theta\\|_2^m\mu_2^m$ is the correct quantity driving the asymptotic behavior.
>
> Last, showing the tightness of the lower bound is much more challenging than we anticipated. (a) Even with the degree parameters given, there is no straightforward testing idea that attains the lower bound. Several tests that work satisfactorily for tensor-SBM suffer from signal cancellation under tensor-DCMM (because tensor-DCMM, which has a large number of free parameters, is more general) and thus lose power. (b) It is hard to estimate the degree parameters when $m\geq 4$. Several estimates that work satisfactorily for $m\leq 3$ have non-negligible biases for $m\geq 4$. We have to propose an iterative algorithm for bias correction (see Section 3.1). No such issue exists for tensor-SBM.
>
> It is a pity that the paper is too compact due to that NIPS has a strict page limit and that we have to present many results to make the story complete. We would like to clarify all the technical difficulties in the revision (by moving some content to the supplement).
>
>
> 2.Why $\pi_i$ appears in the expected degree:
>
> The expectation here is conditional on $\pi_i$. Our degree matching requires ``for fixed $i$ and every possible realization of $\pi_i$, it holds that $\mathbb{E}_1[d_i|\pi_i]=\mathbb{E}_0[d_i]$. We will add a clarification there.
>
>
> 3.An intuitive explanation of how degree matching affects $\chi^2$-divergence:
>
> Great suggestion. We will add an explanation. Intuitively speaking, the $\chi^2$-divergence has many terms, each being the sum (or a function) of terms in a Taylor expansion. Therefore, we can roughly call these terms first-order term, second-order term, and so on and so forth.  When expected node degrees are not matched between the null and the alternative, the first-order term dominates, and correspondingly, a degree-based $\chi^2$-test may have power; see Section 2.4.   When the expected degrees are matched, the first-order term vanishes as we have hoped,   and the degree-based $\chi^2$-test loses power. As a result, the second-order term in the $\chi^2$-divergence now dominates.  The asymptotic behavior of the second-order term is controlled by $\\|\theta\\|_1^{m-2}\\|\theta\\|_2^m \mu_2^m$; thus, it gives a sharp lower bound.
>
>
> 4.Expanding Section 2.4 and Section 3.1:
>
> We are sorry that due to strict page limit, we have to present the two sections in really compact form. We would be glad to expand them in the revision.
>
> In our response to your Point 1, we try to explain the challenges of finding the test in Section 3.1. We hope it is useful.

---

### Author Response · Authors · 2021-08-10
**Why the paper fits NeurIPS**

We thank all reviewers for valuable comments. We are glad that all of them find our paper technically sounding, and especially, some reviewers find our results new, original, important, and that our paper solves a fundamental problem in graphical model.

However, since that there is a strict page limit  and that we have to present many results to make the
story complete, we could not explain the motivation and practical relevance of our problem with more details, and as a result, there are significant misunderstandings on these.  We wish to take a chance to clarify.



Why the paper fits NeurIPS (area, application examples, data, and algorithm):

First, some reviewers think our work grew out of the probability area of random graphs and stochastic block models, and thus our results are only of interest to mathematical probabilist. We do not agree, for our problem grew out of the area of social network analysis, which is a well-accepted and indispensable area in machine learning (e.g., Peter Bickel and Bin Yu at UC-Berkeley, and Stephen Fienberg and Eric Xing at CMU used to work in this area; all these people are well-known in the machine learning community). Also,  in recent years, there are many NeurIPS papers discussing network community detection and related problems, with many interesting applications. The authors have been working in network analysis for many years and the current paper grew out from this area.

In fact, our paper points out that probabilistic random graphs and stochastic block models are sometimes too idealistic to be useful in real applications, so we propose to consider a much broader and more realistic model. Our model extends the well-known Degree-Corrected Block Model (DCBM) by Karrer and Newman (2010) for network analysis to tensor setting and allows for mixed-memberships. It is important to use such a broader model. For example,  in Section 2.4, we show that a test with good powers for the block model setting turns out to be powerless in the broader setting.

Also, as partially explained in Lines 14-16, our paper is directly motivated by a large-scale high-quality data set on the publications of statisticians,  where we have the bibtex (title, author, abstract, keywords, references) and citation data of 83,331 papers in 36 journals, 1975-2015. Since a paper may have $m$ authors,  $m = 1, 2, \ldots$,  the co-authorship relationship is best modeled with a non-uniform hypergraph.  We are interested in (a) how to use the hypergraph to build a community tree (and so to visualize the coauthor topology of statisticians), and (b) how to measure the diversity of an individual author.  Our study can help (a)-(b). For (a), note that a recursive clustering algorithm (like a tree) has many layers, and for each identified community in each layer, a
crucial problem is to decide whether we should  further divide it into several communities or not. This is exactly the global testing problem we study in this paper. For (b), to measure the diversity of an author, we consider his/her personalized (or ego) hypergraph, where each node is a co-author, and we think his/her co-authorship as diverse if and only if his/her personalized hypergraph has more than one community (so he/she co-authors with people from at least two different groups or research areas). Therefore, this is also the global testing problem we discuss. We have done both (a)-(b) in the network setting with the data set mentioned above. However, since this data set is best modeled with a non-uniform tensor, instead of a network (e.g., a paper may have 3 or more authors; it is better to model the co-author relationship as a hyper-edge instead of an edge), we have to attack the problem with a tensor model as in the current paper.

Last, additional to a tight lower bound (which, as all reviewers agree, solves a difficult problem), we also present an upper bound. It is true the algorithm we propose is not polynomial time, but we have the following comments.

(i) We conjecture that unless we put a hard-to-check and unrealistic constraint on the model,
polynomial-time algorithm that is optimal does not exist: only non-polynomial time algorithm
can be optimal in such a broad setting. For example, consider an order-$3$ uniform tensor with $2$ communities,  where (a) two communities have equal sizes, (b) $\theta_1 = \theta_2 = \ldots = \theta_n$, and (c) the two slices of  tensor ${\cal P}$ (each of them is a 2 by 2 matrix)  are

$
\left[
\begin{array}{ll}
.5 + \epsilon, & .5 - \epsilon \\\\
.5 - \epsilon, &  .5 + \epsilon \\\\
\end{array}
\right], \qquad \mbox{and} \qquad
\left[
\begin{array}{ll}
.5 - \epsilon, & .5 + \epsilon \\\\
.5 + \epsilon, &  .5 - \epsilon \\\\
\end{array}
\right],
$

respectively, where $\epsilon$ depends on $n$. Our preliminary study of the computational lower bound suggests that there does not exist a polynomial time algorithm that is optimal (of course, our proposed test is optimal in this case, though it is not polynomial time). On one hand, such a result says our proposed algorithm (though not polynomial time) is practically relevant, as we won't have any polynomial-time algorithm.  On the other hand, if we really need polynomial time algorithm that is optimal, we must put a constraint
on our model. Of course, we wish to have a constraint that is as weak as possible, but how to identify such a constraint is a highly nontrivial problem. This problem has never been discussed before in the literature, and we intend to study it in the near future.

(ii). While our  algorithm is not polynomial time, it does not mean it is easy to construct. In fact, even if we allow algorithms that are not polynomial time, no existing method is optimal and it is especially challenging to construct one when $m \geq 4$ ($m$ is the order of tensor):  to match with the lower bound, we have to develop a new algorithm to estimate $\theta_1, \theta_2, \ldots, \theta_n$.

(iii) At the same time, in Section 3.1, we have proposed a greedy version of our algorithm, which approximates our
testing statistics. The algorithm is fast, so we do have some contribution on this front.

---

### Decision · Program_Chairs · 2021-09-27

**Decision:**

Accept (Poster)

**Comment:**

The paper's main contribution is to provide a crisp theoretical characterization of the feasibility of detecting multiplicity of underlying communities in a degree-corrected mixed membership hypergraph model.

It has been recognized by all reviews that this is a significant achievement that will be of interest to researchers working on stochastic block models and related inference questions. The reviewers did not give very high marks on the basis that the topic may be of interest to only a limited subset of the NeurIPS community, also pointing to the fact that the paper could be made more attractive if a compelling application was brought forward.

The authors' reply to these comments is an argument that the problem they tackle is important in network science, an area which is relevant for NeurIPS, and a description of concrete applications to hypergraphs of co-authorships in scientific articles. The authors further point to the usefulness of their result to hierarchical clustering, and explain in greater detail how their non-polynomial time test informs the design of the polynomial test used in the experiments.

I believe that these answers by the authors alleviate the concerns expressed by the reviewers, and suffice to justify acceptance of the paper to the conference.